# Multi-Temporal Satellite Investigation of gas Flaring in Iraq and Iran: The DAFI Porting on Collection 2 Landsat 8/9 and Sentinel 2A/B

**DOI:** 10.3390/s23125734

**Published:** 2023-06-20

**Authors:** Mariapia Faruolo, Nicola Genzano, Francesco Marchese, Nicola Pergola

**Affiliations:** 1Institute of Methodologies for Environmental Analysis, National Research Council, 85050 Tito Scalo, Italy; francesco.marchese@cnr.it (F.M.); nicola.pergola@cnr.it (N.P.); 2Satellite Application Centre (SAC), Space Technologies and Applications Centre (STAC), 85100 Potenza, Italy; nicola.genzano@unibas.it; 3School of Engineering, University of Basilicata, 85100 Potenza, Italy

**Keywords:** DAFI update, gas flaring investigation, multi-platform satellite dataset, OLI, OLI–2, MSI, daytime infrared radiances, radiative power

## Abstract

The synergic use of satellite data at moderate spatial resolution (i.e., 20–30 m) from the new Collection 2 (C2) Landsat-8/9 (L8/9) Operational Land Imager (OLI) and Sentinel-2 (S2) Multispectral Instrument (MSI) provides a new perspective in the remote sensing applications for gas flaring (GF) identification and monitoring, thanks to a significant improvement in the revisiting time (up to ~3 days). In this study, the daytime approach for gas flaring investigation (DAFI), recently developed for identifying, mapping and monitoring GF sites on a global scale using the L8 infrared radiances, has been ported on a virtual constellation (VC) (formed by C2 L8/9 + S2) to assess its capability in understanding the GF characteristics in the space-time domain. The findings achieved for the regions of Iraq and Iran, ranked at the second and third level among the top 10 gas flaring countries in 2022, demonstrate the reliability of the developed system, with improved levels of accuracy and sensitivity (+52%). As an outcome of this study, a more realistic picture of GF sites and their behavior is achieved. A new step aimed at quantifying the GFs radiative power (RP) has been added in the original DAFI configuration. The preliminary analysis of the daily OLI- and MSI-based RP, provided for all the sites by means of a modified RP formulation, revealed their good matching. An agreement of 90% and 70% between the annual RPs computed in Iraq and Iran and both their gas-flared volumes and carbon dioxide emissions were also recorded. Being that gas flaring is one of the main sources of greenhouse gases (GHG) worldwide, the RP products may concur to infer globally the GHGs GF emissions at finer spatial scales. For the presented achievements, DAFI can be seen as a powerful satellite tool able to automatically assess the gas flaring dimension on a global scale.

## 1. Introduction

There is currently an extensive amount of Earth observation (EO) data, collected by an increasing number of satellites, freely accessible to everyone thanks to the open data policies adopted by most governments and spatial agencies [1,2]. This favors the analysis of environmental phenomena on a global scale and the developments of new methods devoted to better understanding the Earth and its environment [1,3,4]. However, the challenges related to big data, such as volume, velocity and variety, require moving away from traditional local processing and data distribution methods to lower the barriers caused by data size and related complications in data management [5]. To tackle these issues, EO data cubes (EODC) are the new paradigm revolutionizing the way users can interact with EO data and are a promising solution to store, organize, manage and analyze them [6]. Different EODC implementations are currently operational such, as the Google Earth Engine (GEE), a cloud-based geospatial processing platform hosting petabyte scales of over 40 years of raw, remotely-sensed imagery and ready-to-use products, which is useful for large-scale environmental monitoring and analysis [7].

A global phenomenon like the routine gas flaring, occurring in gas processing plants and oil refineries all over the world, is one of the top environmental challenges that must be addressed for improving sustainability measures across the petroleum industry [8,9,10]. The continual emissions of large gas-flared volumes which introduce huge amounts of carbon dioxide and GHG (greenhouse gas) emissions during the GF process is alarming and calls for proper monitoring [11]. Unsurprisingly, the negative GF impact on both the environment and climate is increasingly gaining the attention of researchers, climate change advocates and environmentalists [8]. Data about global gas flaring intensity, obtained from reputable and reliable sources, serve for tracking the progress of its reduction and in delivering the commitments of the Zero Routine Flaring by 2030 and zero emissions by 2050 captured in post-Paris climate policy [10,12,13,14].

EO data, ensuring a continuous long-term flux of multi-spectral/spatial/temporal observations/products, have become essential to acquire independent and global information on gas flaring evolution [4]. In this direction, the GGFR (Global Gas Flaring Reduction), in partnership with the U.S. National Oceanic and Atmospheric Administration (NOAA) and the Colorado School of Mines, is engaged and each day providesa global picture of gas flaring sources, based upon nighttime infrared observations from VIIRS (visible infrared imaging radiometer suite) at 750 m of spatial resolution [15].

Recently, some authors [16,17,18,19,20] demonstrated the power of daytime satellite imagery as well, set at a finer spatial resolution than VIIRS (i.e., 30/20 m), in retrieving detailed information on thermal anomalies, including the ones coming from gas flaring. Currently, the only method working on a multi-platform collection (i.e., Landsat 8 Operational Land Imager (L8 OLI) and Sentinel-2 MultiSpectral Instrument (S2/MSI) imagery) for gas flaring sites identification is that proposed by [18] for Texas. In this paper, we present and discuss an upgrade configuration of the DAFI (daytime approach for gas flaring investigation) approach, recently developed by [20], to detect gas flaring sites on a global scale through the analysis of Collection 1 L8 OLI scenes (i.e., DAFI v1, https://sites.google.com/view/flaringsitesinventory, accessed on 10 January 2023). The DAFI v2, running under GEE, where it intercepts a virtual data cube (hereafter named virtual collection, VC), including a multi-year time series of Collection 2 (C2) L8/L9 OLI and S2 A/B (S2) MSI daytime infrared radiances, has been designed to both detect gas flares and quantify their radiative power (RP).

The main objective of this work was to assess the add-on value of a multi-platform system for gas flaring identification. Jointly, a novel step was added to the DAFI v1 configuration to quantify the radiant heat from the harmonized satellite collection.

The findings provided by the DAFI v2, as the new gas flaring sites database and their RP estimates, are presented and discussed, using Iraq and Iran as testing areas. The latter, in fact, ranked second and third, in 2022, among the most affected countries for gas flaring worldwide [13].

A key point of this paper is to highlight the valuable contribution DAFI may have in the international scenario by providing a global scenario of the areas most affected by gas flaring in terms of both number of sites and their emissive power intensity.

## 2. Regions of Interest

Both land and marine zones (see black and cyan lines in Figure 1) of the Iraq and Iran regions are used as test areas. Performances of DAFI v2 are assessed even by comparison with DAFI v1 achievements (sky-blue dots in Figure 1) in order to evaluate any possible improvements in detection sensitivity and monitoring capability. The DAFI v2 running on the global scale is currently in progress.

As shown in Figure 1, DAFI v1 identified 158 and 165 gas flares in Iraq and Iran, respectively (sky-blue dots in Figure 1). Most of the of Iraqi and Iranian oil and gas fields and infrastructure are gathered and concentrated in specific regions (i.e., along the east side for Iraq, and the southwest part of Iran) (Figure 1), where there are several populated towns. This increases the importance of the flare recovery and utilization facilities in those areas to reduce air pollution and the population’s health risk of the flare gas emission [21,22]. Indeed, both countries are highly affected by gas flaring, as highlighted by Figure 2 and Figure 3.

Based on the latest statistics released by the GGFR, Iraq and Iran hold the second and third position, among the top 10 gas flaring countries, consistently for the last 10 years [13]. Figure 2a shows their significant role within the global GF scenario, with the highest amounts of gas-flared volumes measured in 2021 (i.e., 17.80 BMC and 17.38 BCM), after the Russian Federation peak (i.e., 25.25 BCM) [15]. In addition, these countries recorded the highest change in gas-flared volumes between 2012 and 2021, +5.1 and +6.3 BCM, against an overall global change of +1.3 BCM (Figure 2b). A mostly linear increase in BCM was recorded in years 2012–2021 for Iraq (Figure 2b, green dashed line), where around 40% of the gas production flared [24]. According to a World Bank estimate, Iraq flares around 16 BCM of gas per year, or 0.5% of global production, which has the potential to power three million homes [22]. A more variable growth was observed for Iran, with the most BCM pronounced levels in 2018 and 2021 (Figure 2b, green line). The Word Bank says the ratio of flared gas to produced oil in Iran is 15.36 cubic meters per one barrel, the highest level in the world after Venezuela and Algeria.

As shown in Figure 3, the higher the BCM, the higher the CO_2_ emissions from gas flaring will be.

According to the Global Carbon Atlas’s latest statistics [25], 186 million tons (MtCO_2_) of CO_2_ GHGs were introduced into Iraq’s atmosphere in 2021, 33.5 of which came from gas flaring (Figure 3a). Iran, with its 749 million tons of greenhouse gas emissions, ranked 6th globally in 2021, with only ~4% coming from gas flaring (Figure 3a). Similar to the BCM temporal evolution (Figure 2b), the high levels of GHGs in Iraq and Iran reveal the strong link between the two parameters and the urgent need to develop independent, accurate and reliable methods for gas flaring emissions estimation, especially for evaluating the results of countries’ efforts in reducing/stopping gas flaring in the upcoming years.

## 3. Data and Method

### 3.1. Satellite Data

The joint use of L8 OLI, L9 OLI–2 and S2 MSI data for EO applications is suggested because they are open and freely available, fully interoperable and best suited to monitor large areas [26]. The aforementioned platforms are placed in sun-synchronous orbits and make similar measurements in terms of spectral and spatial characteristics [27].

The DAFI v2 code runs in parallel on daytime infrared radiances acquired by OLI, OLI-2 and MSI. Data characteristics are summarized in Table 1.

The fourth generation of Landsat is composed of two similar sensors with very high spectral and radiometric sensitivities: OLI and OLI–2. OLI, carried aboard Landsat 8, was launched on 11 February 2013, and OLI–2, aboard Landsat 9, was launched on 27 September 2021 [28]. Near infrared (NIR) and short-waver infrared (SWIR) radiances measured by OLIs during 2013–2022, were collected and processed, for a total of 33.911 images (~40 TB) over the ROIs. In particular, we processed the new Collection 2 (C2) level 1 data, released in December 2020 [29], with improved geometric accuracy, radiometric calibration, additional higher-level products and more quality assessment bands as well as updated and consistent metadata files and a cloud-optimized file format. The area covered by each scene was 185 × 180 km, which, for the study cases, corresponded to 134 tiles [30], each one with a 185 km swath. The temporal resolution was 16 days for L8, reduced to 8 days as far as both satellites were considered together (i.e., L8 + L9).

Moreover, we analyzed NIR and SWIR radiances from the MSI, on board the S2A and 2B satellites (operational since 23 June 2015 and 7 March 2017, respectively), acquired during 2015–2022. A total of 302 S2 tiles, each one covering a swath of 290 km, for a total of 245.242 images (~196 TB) were collected over the ROIs. The Sentinel 2A revisit period was 10 days. Fully operational in combination with Sentinel 2B, a 5-day global coverage was achieved [31]. In particular, we processed the harmonized Sentinel 2 Level 1C (L1C) products released after January 2022, which take in account the last upgrades and correct the known errors in the processing chain [32].

In addition to the spectral bands, the cloud cover information was extracted for both collections. For L8 and L9, the Level 1 quality assessment band (QA_PIXEL band), a bitmask band with cloud mask information, was used to automatically flag cloud-free pixels. For S2, the “COPERNICUS/S2_CLOUD_PROBABILITY” Image Collection, at 10 m, was first resampled at 20 m and then integrated within the S2-L1C datasets. A cloud probability (i.e., probability that the pixel is cloudy) threshold of 5% was set to select cloud-free pixel.

### 3.2. The DAFI v2 Approach

In Figure 4, we show the flowchart of DAFI v2, highlighting the procedural rules executed in parallel on the two satellite collections (i.e., C2 L8/9 OLI and S2 MSI). Despite the original configuration (i.e., DAFI v1), here the thermal characterization of the GF hotspots was implemented and added as an additional output (see panel B in Figure 4) and discussed in the following.

A combination of the NHI (normalized hotspot indices), which were proposed for hot source identification and discussed in detail in several previous studies [33,34,35] and computed from the NIR and SWIR bands, are used by the DAFI v2 (similar to DAFI v1, [20]) to select the hot pixels (see panel A, Line A.1 in Figure 4a). A gas flare corresponds to a pixel with a NHI_SWNIR_ value greater than 0 or saturated at 1.6 μm (named “Extreme Pixel”) [20,34]. The time persistence of these signals enables the discrimination of a gas flaring-affected pixels (i.e., occurrence frequency greater or equal to 10%) from other sources having a similar spectral behavior (e.g., fires) but different temporal distribution (i.e., occasional, seasonal, non-systematic) (see panel A, Line A.2 in Figure 4a). This procedure selects, among all possible hot sources, those characterized by a temperature above ~1600 K, emitting strongly in the 1.6 μm band.

Each detected GF was then classified in a qualitative and quantitative manner (see panel B, in Figure 4b).

In the first case, since a long-term persistence class was assigned to each detected GF, the latter was classified depending on the estimated OF value (see panel B, Line B.1 in Figure 4b). An indication about the time, expressed as the number of years and months, during which the gas flaring was practiced, can be also obtained.

Concerning the GF quantitative characterization, the emissive power of the hot source was assessed by computing the radiative power, using the *SWIR*1 (1.6 µm) radiances. In this work, the daily RP (RP_d_) is quantified using a modified version of the formulation proposed by [36] (see (Equation (4) in [36]) (see panel B, Line B.2 in Figure 4b):(1)$$RP_{d}=\sum_{i}\left(SWIR1_{HS}-\bar{SWIR1_{BG,m}}\right)\times A\times \frac{\sigma }{a}\times 10^{-6}\;[\mathrm{MW}],$$
where:-*SWIR*1*_HS_* is the radiance measured at 1.6 μm in correspondence of the pixel *i* flagged as hotspot (*HS*) within the *HS* buffer (W∙m^−2^∙sr^−1^∙μm^−1^);-$\bar{SWIR1_{BG,m}}$ is the *SWIR*1 radiance of the background. This value, previously determined on a monthly scale, corresponds to the spatial mean of the monthly temporal means computed over all the clear and not hot pixels included in the background (*BG*) buffer (W∙m^−2^∙sr^−1^∙μm^−1^);-*A* is the area of the pixel (m^2^);-*σ/a* is the RP coefficient that maps from the spectral radiance to radiant emittance (sr∙μm).

For OLI, the *σ/a* RP coefficient, derived from the sensor spectral response function is equal to 7.33 sr µm (see Table 1 in [36]. Since this value is similar to VIIRS, regardless of spatial resolution (i.e., 7.32 sr∙μm) [36], we assumed the same RP coefficient also for the MSI to compute the radiative power of the gas flaring sources.

The main novelty we introduced in Equation (1), beside the porting of the RP coefficient from OLI to MSI, was the way the hotspot and background *SWIR*1 radiances are computed. Indeed, two buffers, with a radius size chosen after testing several values in the investigated countries, were used for this purpose (see panel B, Line B.2 in Figure 4b). In future studies, the possible influence of these sizes on the RP estimates will be better investigated.

For the hotspot, a buffer of 70 m surrounding each gas flare centroid (pink circle in Figure 4b, panel B, Line B.2) is the most suited to include all the GF-affected pixels. For each image acquired over the area in the investigated years, pixels flagged as hotspots according to the DAFI prescriptions (see panel A in Figure 4a) are selected and used, one by one, for the RP computation (see panel B, Line B.2 in Figure 4b).

For the background, a 300 m buffer close to the GF location (depicted in blue in Figure 4b, panel B, Line B.2) seem to assure an accurate spectral characterization of the non-gas-flaring region. To define the BG behavior at 1.6 μm, we applied the flux depicted in the figure below.

First, we filtered, on each monthly image available over the ROIs, all clear and no hot pixels (*j*, see the first block in Figure 5). To minimize the impact of clouds on the background selection, we computed, for each pixel *j* previously selected within the BG buffer, the monthly mean of the *SWIR*1 radiance (i.e., $\bar{SWIR1_{j,m}}$, see the second block in Figure 5). Then, a spatial mean within the BG buffer was calculated for each month (i.e., $\bar{SWIR1_{BG,m}}$, see the third block in Figure 5). The RP_d_ is the sum of all single RPs computed for the thermal anomalies (i.e., NHI_SWNIR_ > 0 *OR* extreme pixel) flagged over the single acquisition, within the HS buffer, as the difference between the *SWIR*1*_HS_* and *SWIR*1_BG_ contributions.

## 4. Results

In the following, we show the new GFs map derived by implementing DAFI on the VC dataset (i.e., DAFI_v2) and the results of the radiative power estimations from the detected GFs. The accuracy and reliability of the performed analyses is also assessed through comparison with independent information (e.g., BCM database).

### 4.1. DAFI v2 Gas-Flaring Hotspots

The DAFI porting on the virtual collection produced a new map with about 520 GF sites (i.e., ~52% more than the DAFI v1), 249 in Iraq and 271 in Iran. As described in [20], each GF site is represented by a centroid that is the result of the OF raster map vectorization.

Figure 6 shows the spatial localization of the GF sites identified by the DAFI v2 (pink dots) in Iraq (Figure 6a) and Iran (Figure 6b) compared to the previous detections (DAFI v1, sky-blue squares). The visual inspection of the new GFs, performed through the high-spatial resolution images available in Google Earth, revealed the absence of false identifications in the regions of interest.

Figure 6 shows the improved performances offered by DAFI v2, which was capable of detecting a higher number of real active gas flaring sites than DAFI v1 (i.e., 343). On the other hand, DAFI v2 did not flag three gas flaring sites located in Iran (see white box in Figure 6b). The reason is that the new OF values, measured in a wider temporal window (i.e., up to 2022), were lower than the used 10% threshold (see OF values on the left side of Figure 7), revealing a temporal decline in the GF activity, which is better emphasized by the charts reported below (see Figure 7).

The three gas flares have strongly reduced their activity intensity in 2022. In detail, the GF_3 is not more operational in 2022, for both OLI and MSI findings. The GF_1 activity decreased in the last investigated year of ~70% for OLI and ~50% for MSI. The GF_2 stopped its process in 2022 according to OLI, while heavily dampen its operation (~80%) for MSI.

The new findings provided by DAFI v2 are summarized in Table 2.

In detail, 308 of the new 520 GFs were commonly detected by OLI- and MSI-based collections (Table 2). In this case, we assumed that a GF site belonging to Landsat or Sentinel 2 dataset indicated the same location if the distance between their corresponding centroids was lower than 70 m.

The contribution of each single satellite collection is quantified in the second and third lines of Table 2, where 49 GFs were recognized only by L8/L9 and 163 only by S2, respectively. In general, a very similar number of GF sites has been recognized in Iraq and Iran by both satellite systems, with a contribution of MSI three times greater than the OLI one.

An example of detected gas flares belonging to these sub-classes of Table 2 (i.e., single and virtual collection) is shown in Figure 8.

The gas flare shown in Figure 8a is a detection common to the two collections. The OF equal to 30.9% and 53.5%, for OLI and MSI, respectively, along with the temporal evolution of this occurrence disaggregated by year (see the chart depicted in Figure 8a on the bottom), provide the thermal history of this site. It has been continuously operational since 2013, with variable intensity levels during the years. In addition, as revealed by the maps, showing the thermal anomalies related to the identified GF (i.e., pixels with OF ≥ 10%, on the top of Figure 8a), most of the HS buffer area (white circle in Figure 8a) is covered by thermal anomalies, mainly showing mid-high OF levels (pink- and purple-colored pixels).

Figure 8b,c focuses on two gas flares detected by the single sensor. In both cases, the gas flares belong to the category of “low long-term persistence level” (see classes in the panel A, Line A.1 in Figure 4a).

The gas flare shown in Figure 8b interrupted its thermal activity in 2016, following the OLI-based findings. It is interesting to note that also the MSI ones recognized short activity of this plant, in years 2016–2017, not long enough to reach a 10% in the long-term OF analysis.

The hot pixel in Figure 8c refers to a GF site recognized as such by the MSI-based DAFI v2. A part of a stop in 2019, it is still in operation. For this place, no thermal anomaly was identified in the last 10 years by the OLI configuration.

An updated scenario about the long-term persistence of the gas flares has been provided by DAFI v2 for Iraq and Iran. As detailed in [20], each GF can be classified according to its long-term temporal persistence: the higher the persistence OF, the longer the gas flaring process activity during the investigated temporal window. Figure 9 puts in evidence for this classification for the DAFI v1 and v2 detections (depicted in sky-blue and pink, respectively).

By focusing on Figure 9, despite a parity level of mid-low and mid-high classes (for both DAFI versions, ~20%), the DAFI v2 reverses the percentages of the extreme classes (i.e., high and low), revealing a prevalence of GF facilities working with more continuity in time.

### 4.2. Gas-Flaring OLI- and MSI–Based Radiative Power Estimates

In the following, the RP_d_ retrievals performed through Equation (1) are discussed.

First, the daily RPs computed using OLI and MSI (RP_OLI_ and RP_MSI_) for all common GFs (i.e., 308) are compared to assess the coherence between the estimates coming from different sensors. We selected the coincident acquisitions for this purpose (see Figure 10).

The scatterplots in Figure 10, where the black dots refer to Iraq (Figure 10a) and Iran (Figure 10b) RP estimates, highlight good agreement (R = ~70%) in both countries of the daily RP values retrieved from MSI and OLI data, which were mostly in the range 0–6/7 MW.

Figure 11 shows an example of RPs values retrieved from the temporally close OLI and MSI observations in both onshore (Figure 11a) and offshore (Figure 11b) conditions, within the 70 m radius buffer around the GF. The hot pixels, flagged on satellite scenes from 8 December 2020 (onshore site, at lon 50°38′9″ E, lat 30°2′27″ N) and 14 April 2020 (offshore site, at lon 52°8′9″ E, lat 26°43′12″ N), are depicted in different colors based on the relative RP value. For the marine platform, the false color composite (R = *SWIR*2, G = NIR, B = RED) imagery is displayed in the background to emphasize the gas-flaring platform (Figure 11b), while the GE satellite imagery is uploaded in the case of onshore gas flares. Both investigated gas flares are localized in the Iranian territory.

The sum of the single RPs computed for each hot pixel within the GF buffer defines the characteristics of the gas-flaring source. In more detail, the higher the RP values, the stronger the emissive power of the source. Figure 11 shows a good match between RP_OLI_ and RP_MSI_ values computed over the same selected targets and days of observation, independently from the sensor data (~2.5 MW for the onshore site and ~0.6 MW for the offshore one). The slightly higher RP_MSI_ values, retrieved over both the onshore and offshore sites, may be associated with the higher spatial resolution of MSI data, which enables a more accurate mapping of the gas flares characterized by less intense thermal emissions.

Figure 12 shows the temporal trend of the RPs computed from OLI (since 2013) and MSI (since 2015) data, for the same GFs of Figure 11. In more detail, the plots of Figure 12, including the RPs retrieved until 2022 when all the sensors were operational, show that the OLI and MSI observations were mostly complementary in the time domain and in some cases coincident, allowing for a more continuous quantification of the radiative power. Moreover, the similarity of the RP values demonstrates the relevance of the used multi-sensor system in characterizing both onshore and offshore GFs even in the time domain.

This result assumes a great relevance if the RP role in the gas flaring assessment is considered.

All the RP_OLI_ and RP_MSI_ values, computed in the analyzed temporal window for each detected GF, were summed on an annual basis, at the country level (Iran and Iraq), and then compared with the annual gas-flared volumes (in BCM, provided by the EOG) [15] and CO_2_ GF emissions (in MtCO_2_, provided by the Global Carbon Atlas [25] (see Figure 13).

A good correlation between the RP estimates and the GF emissions, as both volumes and GHG emissions, is evident for both Iraq (R = 70%) and Iran (R = 90%). Based on this finding, the radiative power, retrieved from OLI, OLI–2 and MSI data through the formulation proposed in this paper, will be exploited to quantify the gas-flaring emissions at the country level.

## 5. Discussion and Conclusions

The DAFI v1 method was designed to detect gas-flaring sites, globally distributed, through the automatic processing of L8 OLI daytime satellite infrared data at 30 m spatial resolution. Gas-flaring sources are discriminated by filtering a 10% threshold to a couple of signals in the long period.

In this paper, we have explored the DAFI potential in providing information about the gas-flaring sources when exported on a multi-platform system, integrating OLI, OLI-2 and MSI observations (i.e., the virtual collection). The application of DAFI v2 in Iran and Iraq improved by about 52% the number of gas flares previously identified through DAFI v1. This outcome probably depends on the longer and denser time series which allows for a more complete sampling of the source signals. In addition, comparing the OLI-2- and MSI detection findings highlighted a major contribution offered by the MSI-based DAFI configuration, about three times higher than the OLI one. This benefit is double because, on one side, new GF sites, undetected by both OLI and OLI-2, have been identified, and, on the other side, a more accurate discrimination of multiple gas flares belonging to the same industrial plant seems to be guaranteed. It is well known that gas flares show a similar spectral signature in the SWIR band [37], with a peak around 1.6 μm [31,38]. Nonetheless, when observed at different spatial resolutions, the signal response changes. For this reason, a general higher NHI indices sensitivity has been observed when implemented on MSI data, although almost identical to the spectral range of OLI and MSI NIR/*SWIR*1 bands (see Table 1). This may be due to the higher spatial and temporal resolution of S2-MSI data, enabling an improvement in detection sensitivity which is reflected in the identification of both a higher number of hot pixels and the less intense thermal anomalies.

Another important outcome of this work is the quantitative characterization, in terms of radiative power, of the gas-flaring sources by daytime high spatial resolution imagery. The agreement between the OLI– and MSI–based daily RP estimates, provided by the suggested modified version of the RP equation, encourages the use of a multi-platform satellite system, the first case suggested in the inherent literature, to more accurately assess the GF emissive power in daytime conditions. Furthermore, the good correlation found between RP and GF emissions, both in terms of gas-flared volumes and CO_2_ emissions, suggests the use of RP estimations as proxy for the air quality pollution from the gas-flaring sources. Indeed, a higher emissive power of the source should correspond to a more significant impact of the GF on the surrounding areas.

The synergic use of OLI, OLI-2 and the MSI data, thanks to the increased temporal frequency of observation (up to 2 days) [26,38], opens new perspectives for the monitoring of the gas-flaring sources at local, regional and global scales. Those data are, in fact, spectrally compatible, open and freely available, and suited to be integrated in a satellite-based system implementing the DAFI method. Its code, based on the analysis of the virtual collection, with a volume of ~240 TB, has run in GEE, where the big data were processed by tiles with a cascade running of the algorithm. Such a strategy enabled the obtainment of the gas-flaring products in two weeks. However, as widely discussed in [20], DAFI provides a partial view of the global gas-flaring sites due to the use of a fixed detection threshold, which is more biased towards intense and more persistent gas-glaring sources. Machine learning/deep learning approaches can represent an improvement in this regard and will then be investigated in the future to assess their possible contribution in gas flare identification and characterization.

Our ongoing research is moving in these directions:-Applying the DAFI v2 to the entire globe, for updating the current daytime GF map (https://sites.google.com/view/flaringsitesinventory, accessed on 20 December 2022);-Computing the radiative power of all new detected gas flares to develop a country-based model capable of retrieving the gas-flared volumes and the corresponding CO_2_ emissions from satellite observation only.

Although further improvements of the developed system are possible, DAFI is, to our knowledge, the only scheme operating at a global scale, which automatically analyzes the gas-flaring dimension by means of daytime satellite data.

## Figures and Tables

**Figure 1 sensors-23-05734-f001:**
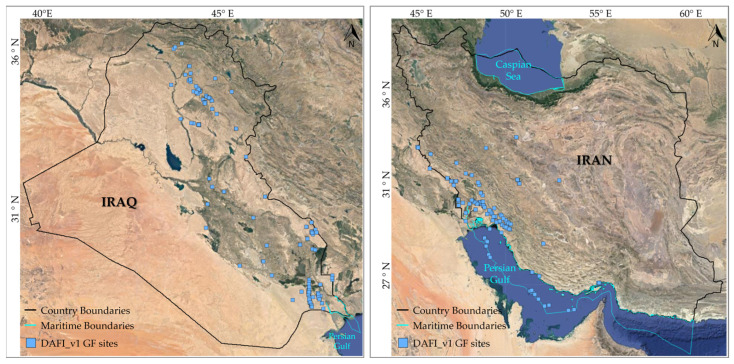
Location map of the regions of interest (ROIs). The sky-blue dots represent the GF sites detected by the DAFI_v1. Land and marine boundaries are represented by black and cyan lines, respectively.

**Figure 2 sensors-23-05734-f002:**
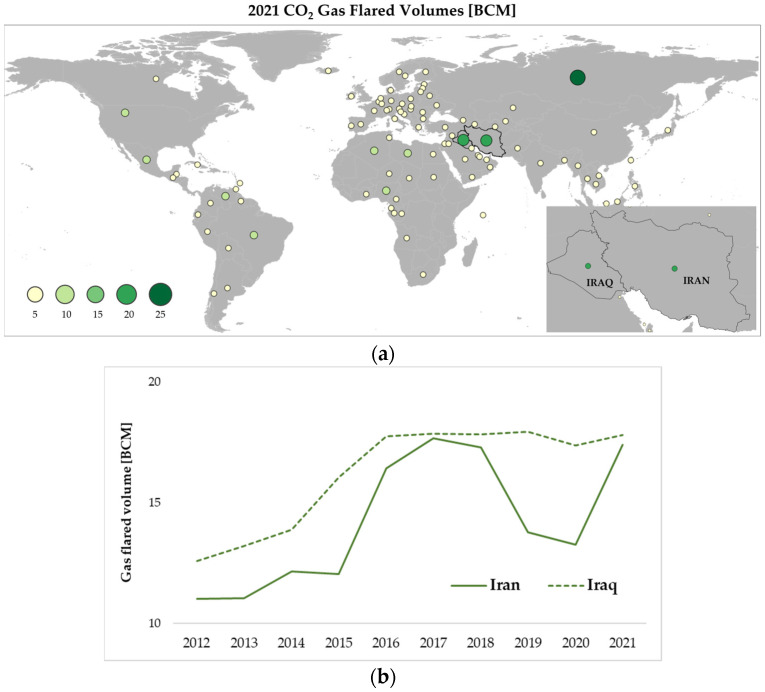
(**a**) Global gas-flared volume (BCM) in 2021 (source: adapted from VIIRS Global Flaring Annual 2012–2021 long [15,23]); (**b**) Temporal trend of BCM in Iraq (dotted green line) and Iran (green line) in years 2012–2021 Adapted with permission from [13].

**Figure 3 sensors-23-05734-f003:**
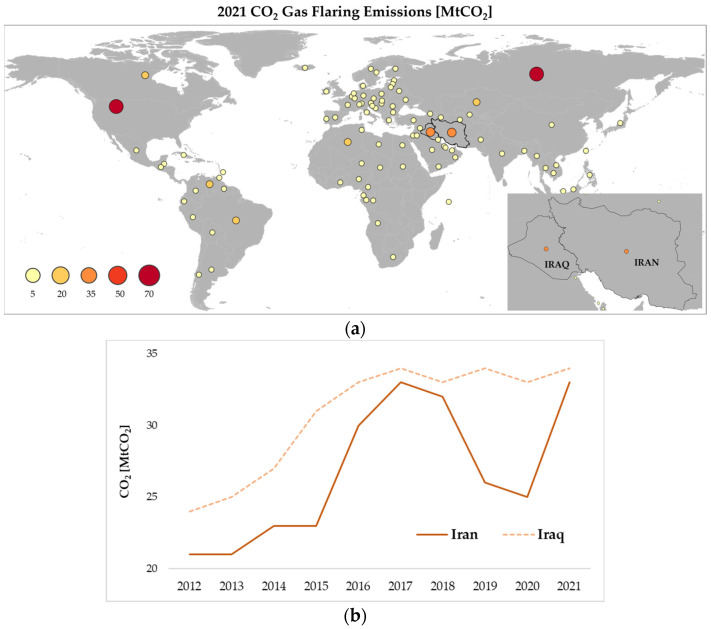
(**a**) Global CO_2_ gas flaring emissions (MTCO_2_) in 2021; (**b**) Temporal trend of CO_2_ emissions from gas flaring in Iraq (dotted orange line) and Iran (orange line) in years 2012–2021. Source adapted from Global Carbon Atlas.

**Figure 4 sensors-23-05734-f004:**
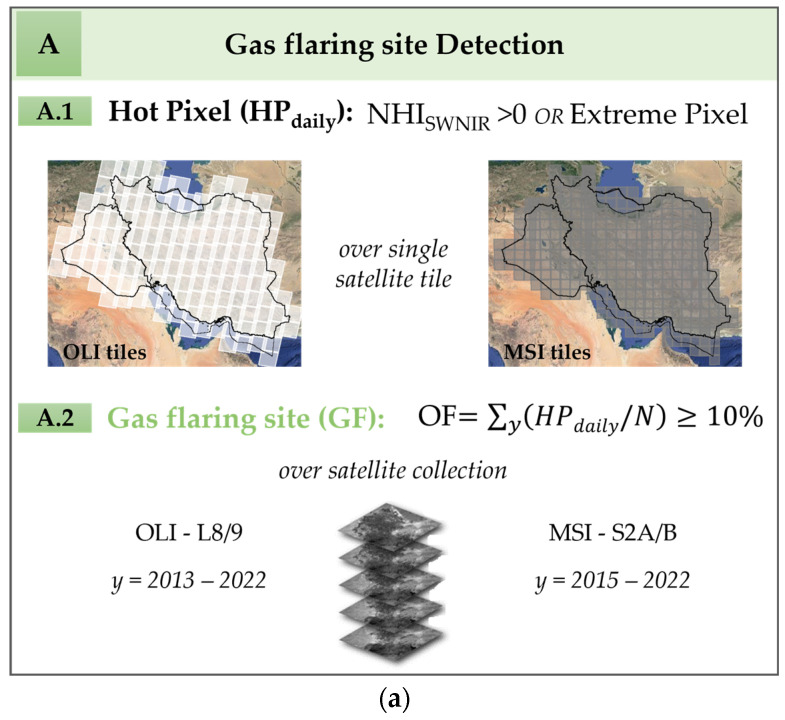
The DAFI v2 methodological steps for (**a**) detecting and (**b**) characterizing the gas flaring sites.

**Figure 5 sensors-23-05734-f005:**
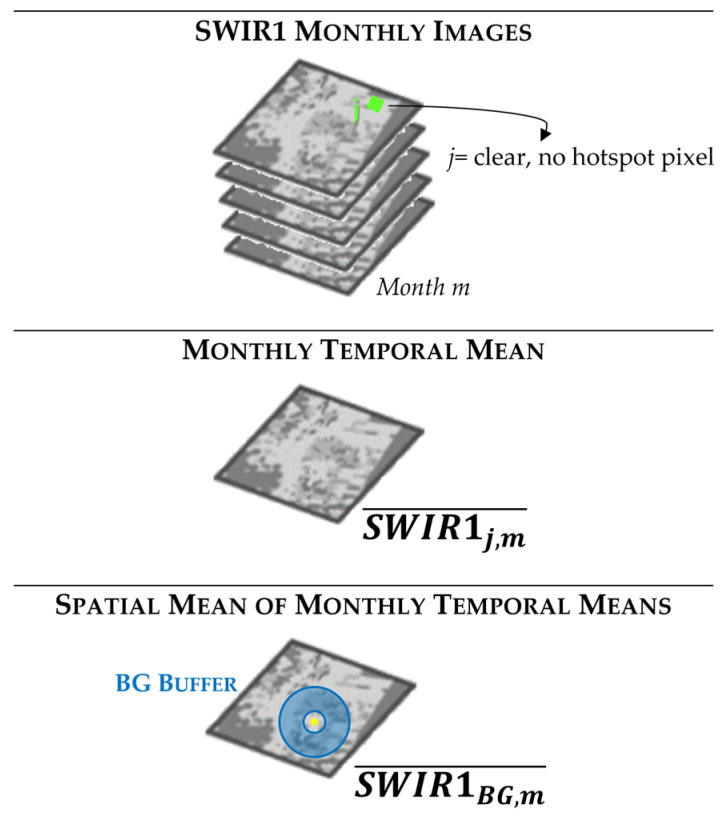
Background characterization approach.

**Figure 6 sensors-23-05734-f006:**
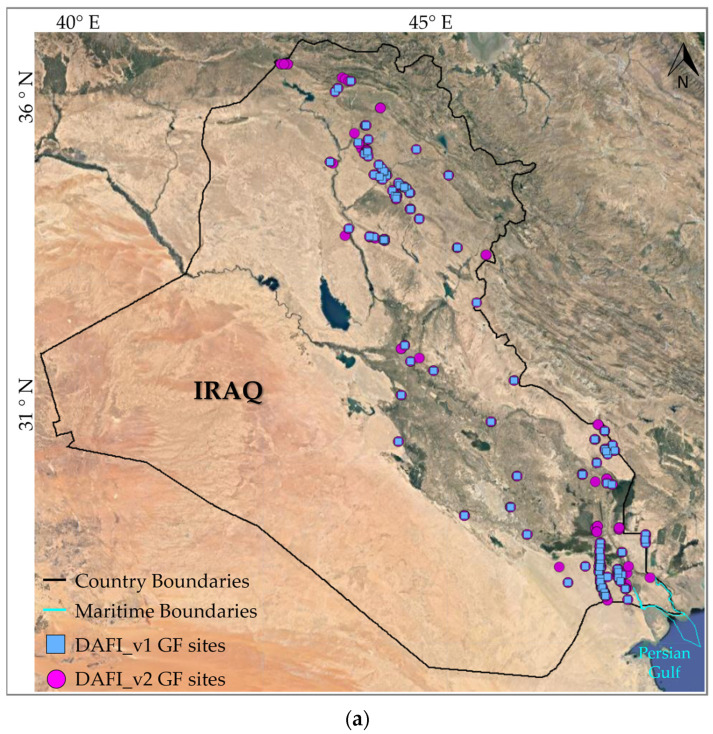
Spatial distribution of GFs detected by DAFI v1 (sky-blue squares) and DAFI v2 (pink dots) in (**a**) Iraq and (**b**) Iran. The three gas flares (GF_1, GF_2, GF_3) within the white box are analyzed in the work.

**Figure 7 sensors-23-05734-f007:**
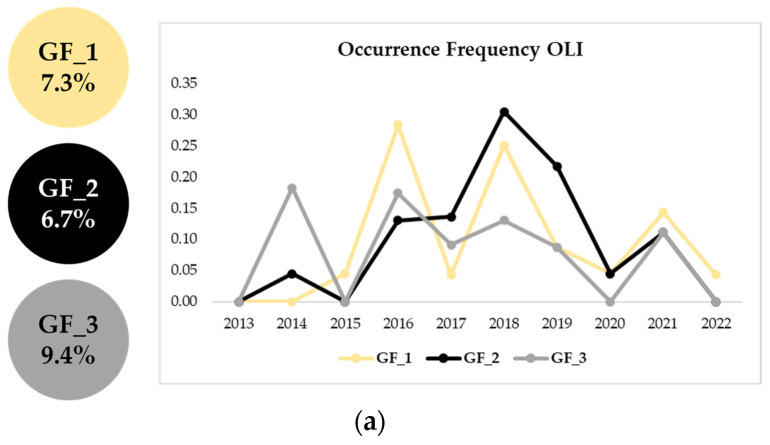
Annual occurrence frequencies computed for the three gas flares highlighted within the white box in Figure 6b, which were not identified by DAFI v2 (OF < 10%), using both (**a**) OLI and (**b**) MSI data.

**Figure 8 sensors-23-05734-f008:**
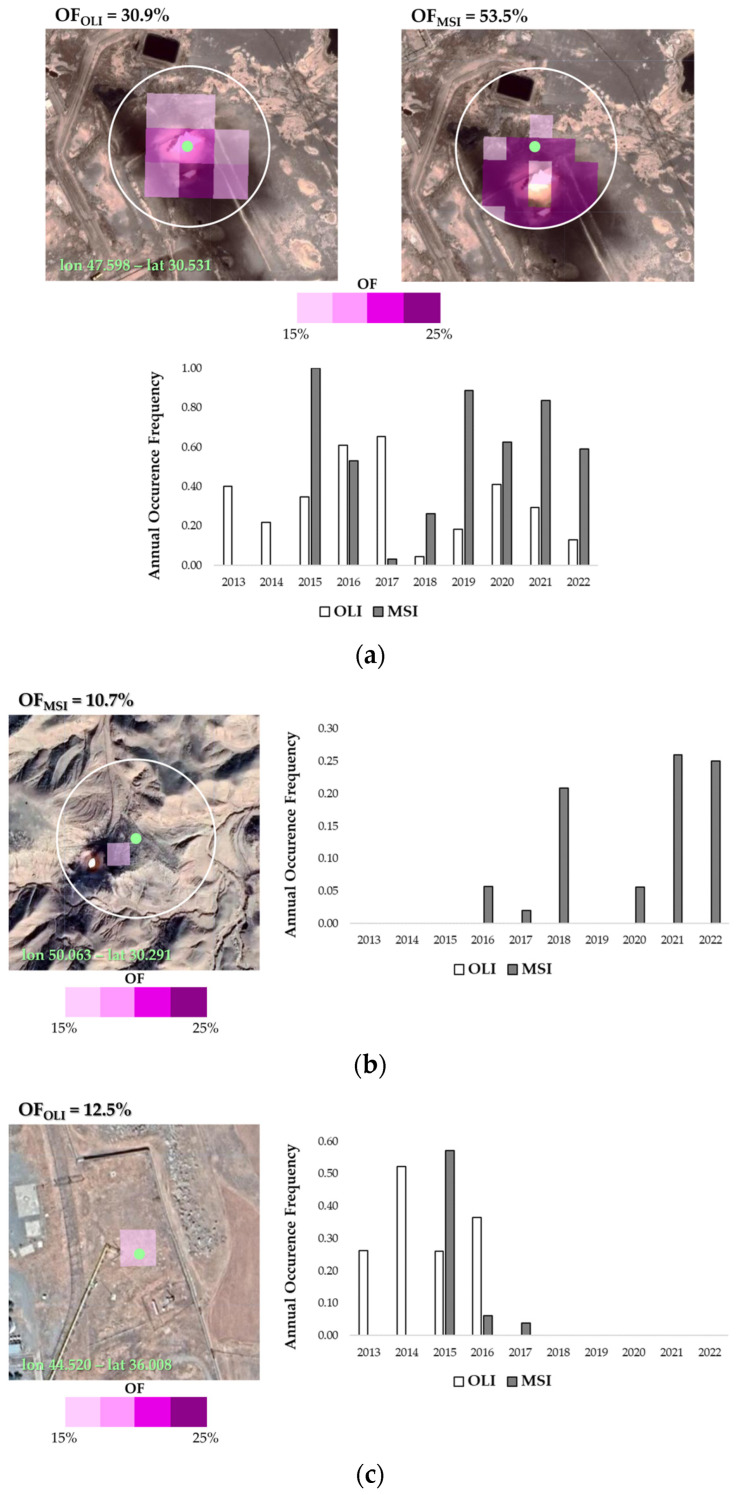
Gas flare detected by (**a**) the combined sensors and the single (**b**) OLI and (**c**) MSI sensor. For each site, the annual occurrence frequencies computed using OLI (OF_OLI_) and MSI (OF_MSI_) data, are also reported.

**Figure 9 sensors-23-05734-f009:**
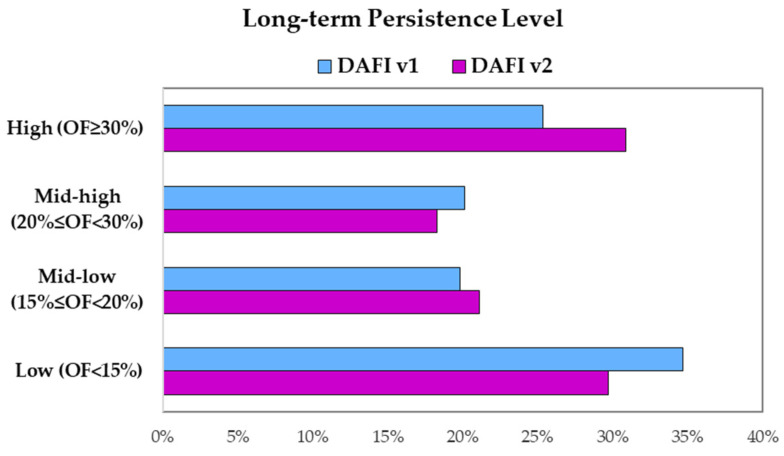
Persistence classes (from low to high) distribution for DAFI v1 (sky-blue bars) and DAFI v2 (pink bars).

**Figure 10 sensors-23-05734-f010:**
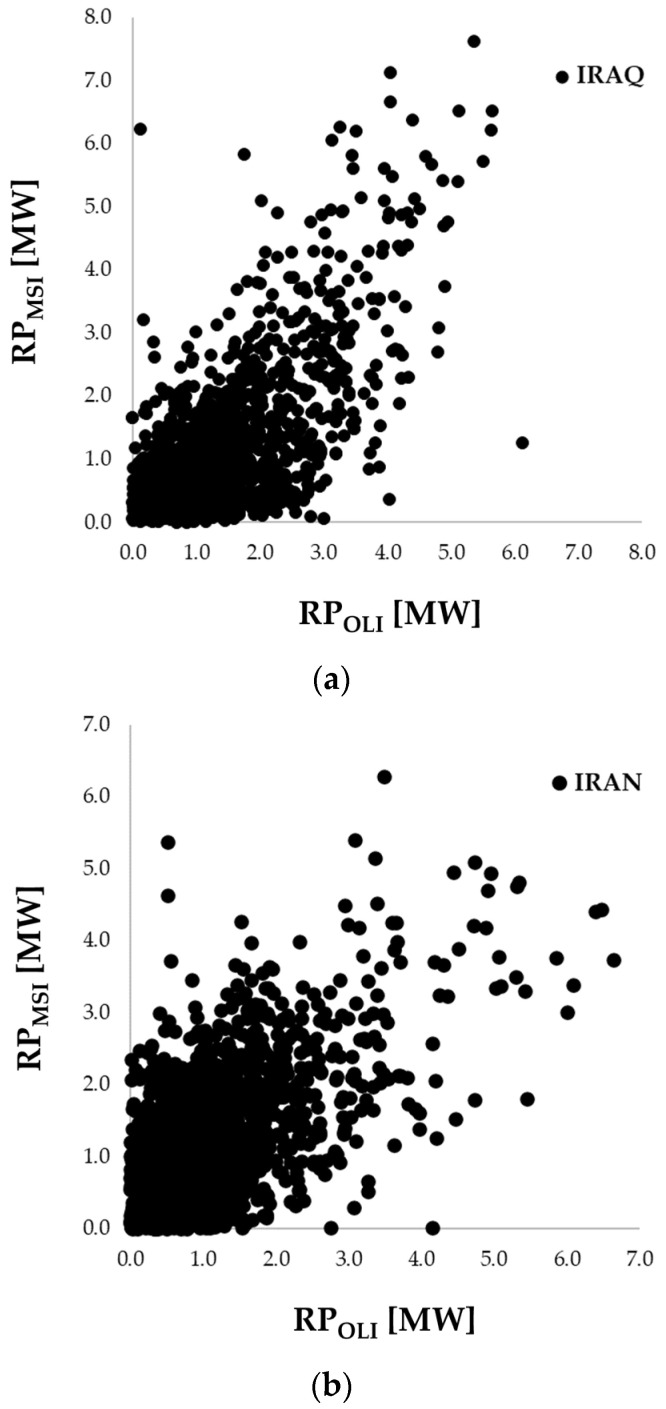
Scatterplot of RP_d_ estimates derived from OLI (*x*-axis) and MSI (*y*-axis) datasets for (**a**) Iraq and (**b**) Iran.

**Figure 11 sensors-23-05734-f011:**
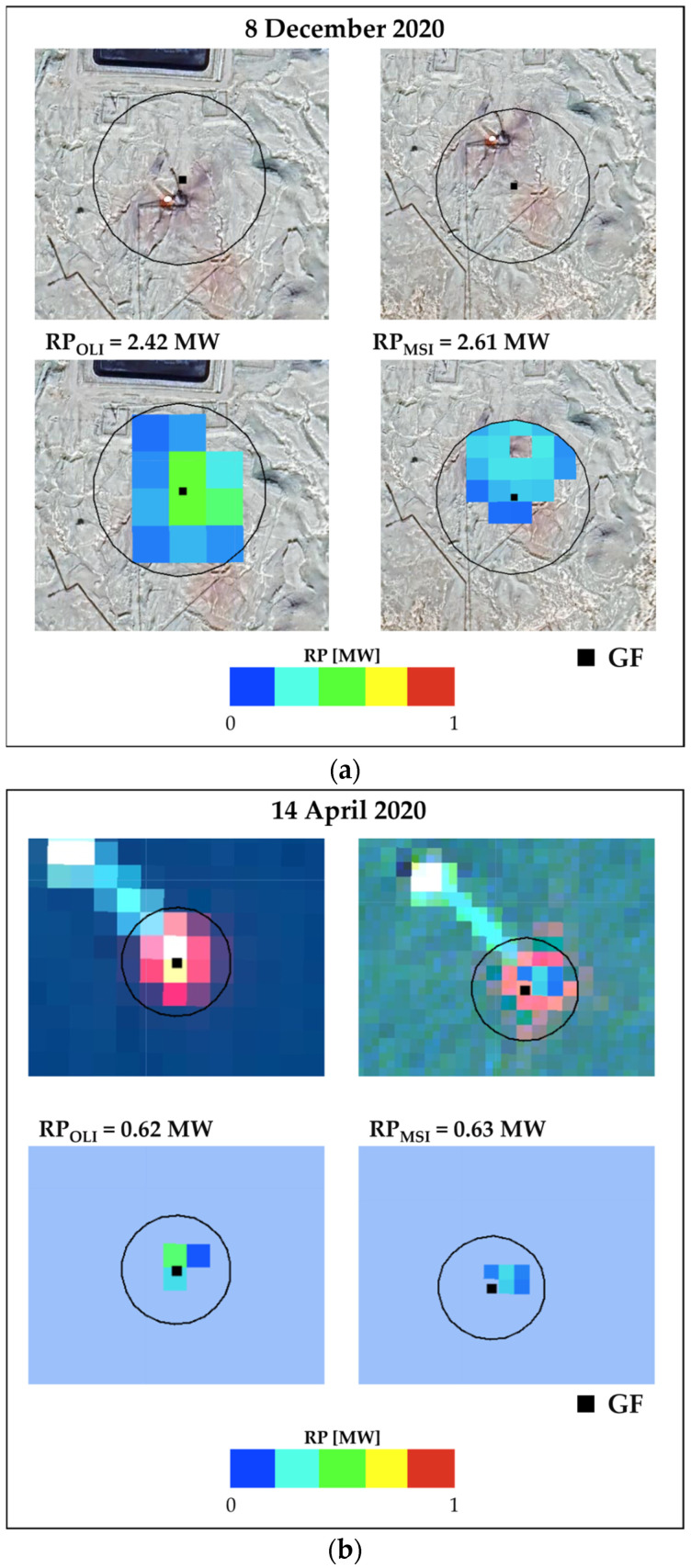
The daily RP estimates (based on OLI, on the left side and MSI, on the right side) from DAFI v2 for the GF site detected in the (**a**) onshore (path 164, row 039 of L8; tile T39RVP of S2) and (**b**) offshore (path 162, row 041 of L8; tile T39RXK of S2) Iranian territory. In background, the false color composites (*SWIR*2, NIR, RED) of each image for the offshore site and the GE satellite images for the onshore one have been uploaded.

**Figure 12 sensors-23-05734-f012:**
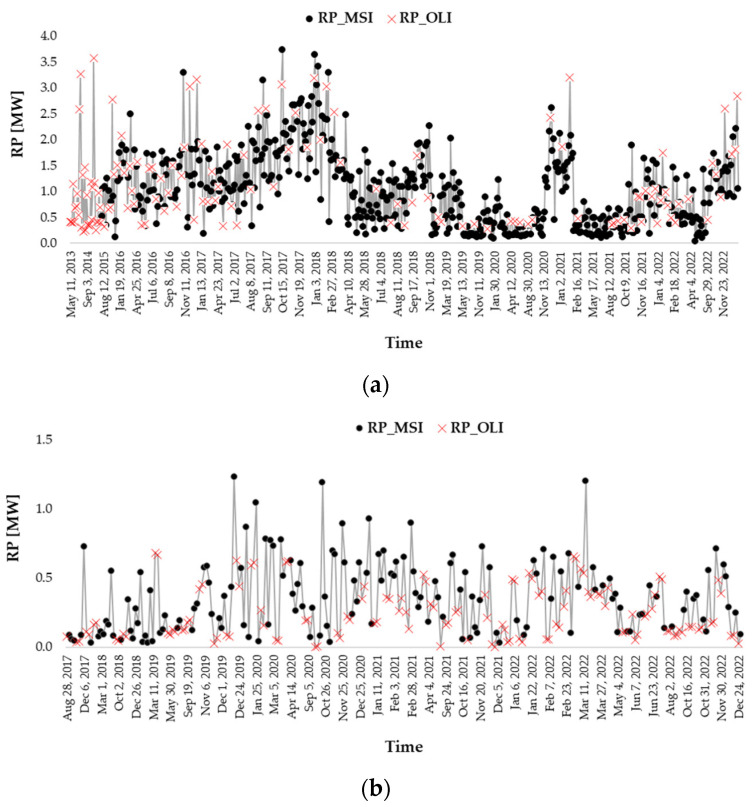
Temporal trend of the daily RP values, computed from the OLI (red crosses) and MSI (black dots) collections for the (**a**) onshore and (**b**) offshore Iranian sites.

**Figure 13 sensors-23-05734-f013:**
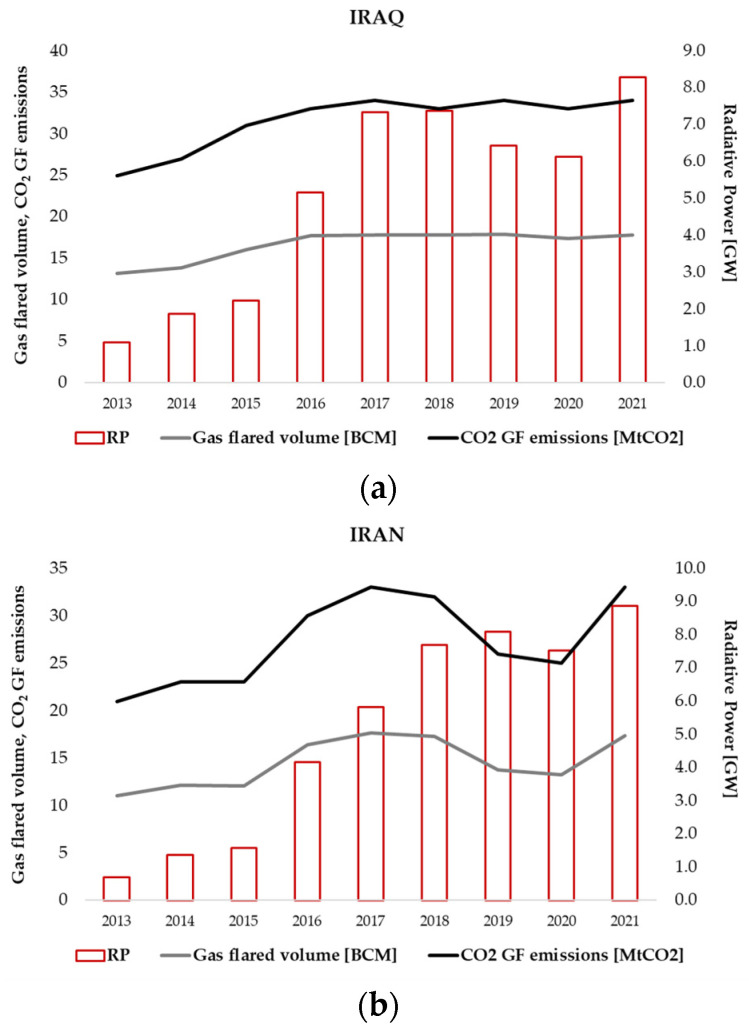
Annual RP values (purple histogram), computed on the detected GFs, compared with the gas-flared volumes (black line) and the CO_2_ GF emissions (gray line) annually recorded in (**a**) Iraq and (**b**) Iran.

**Table 1 sensors-23-05734-t001:** Characteristics of Landsat OLI and Sentinel-2 MSI bands used in this study.

VC Collection	OLI/OLI–2	MSI
**Period**	04/2013–12/2022 (L8) 09/2021–12/2022 (L9)	06/2015–12/2022 (S2A) 03/2017–12/2022 (S2B)
**Band/Central wavelength (μm)**	B1 (0.443), B5 (0.865), B6 (1.610)	B5 (0.705), B8A (0.865), B11 (1.610), B12 (2.190)
	QA_PIXEL	probability
**Spatial** **resolution** **(m)**	30	20
		10
**Temporal** **resolution (days)**	16 (single satellite) 8 (constellation)	10 (single satellite) 5 (constellation)
**Provider**	USGS/NASA Landsat Program	EU/ESA Copernicus Program

**Table 2 sensors-23-05734-t002:** GF detections provided by DAFI v2.

Collection Source	IRAQ	IRAN	Total
**L8/L9 & S2**	141	167	308
**L8/L9**	25	24	49
**S2**	83	80	163
**VC**	249	271	**520**

## Data Availability

L8-OLI and S2-MSI data analyzed in this study are available under the Google Earth Engine platform.

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
