# Peer review of "Multi-Temporal Satellite Investigation of gas Flaring in Iraq and Iran: The DAFI Porting on Collection 2 Landsat 8/9 and Sentinel 2A/B"

_sensors, 2023, doi:10.3390/s23125734_

Round 1

Reviewer 1 Report

In this paper authors have presented the synergic use of satellite data at moderate spatial resolution, from the new Collection 2  Landsat-8/9 Operational Land Imager and Sentinel-2  Multi- spectral Instrument, which provides a new perspective in the remote sensing applications for gas flaring identification and monitoring, thanks to a significant improvement in the revisiting time (up to ~3 days). In their study, the Daytime Approach for Gas Flaring Investigation, recently developed for identifying, mapping and monitoring gas flaring sites at global scale, using the L8 infrared radiances, has been ported on a virtual constellation to assess its capability in understanding the gas flaring characteristics in the space-time domain. 

The paper is very intresting.

I would like authors to explain how they "treated" big data set for data processing, because of its volume, velocity and variety.

Is "A cloud probability (i.e., probability that the pixel is cloudy) threshold of 5% was set to select cloud free images" means that they took only 5% of data for processing? 

What was the method of optimiazation this data?

Author Response

Please, see the attachement.

Reviewer 2 Report

This study assessed the added value of multi-platform system for gas flaring detection. One step to quantify the radiant heat is added to the DAFI system. The results are promising, and the manuscript is well organized. I suggest to accept it after one minor rvision. 

Minor:

Please add the description of the added new step to quantify the radiant heat in the abstract and its related impacts. 

Reviewer 3 Report

Dear Authors

After detailed readings in the manuscript, entitled: “Satellite multi-temporal investigation of gas flaring in Iraq and Iran: the DAFI porting on Collection 2 Landsat 8/9 and Sentinel 2A/B”. By performing procedures from the new Collection 2 (C2) Landsat-8/9 (L8/9) Operational Land Imager (OLI) and Sentinel-2 (S2) Multispectral Instrument (MSI), provides a new perspective in the remote sensing applications for gas flaring (GF) identification and monitoring, thanks to a significant improvement in the revisiting time. Thus, the innovative character of the manuscript can be seen, I believe it will arouse the curiosity of the journal readers. I suggest ACCEPT the manuscript with minor corrections:

1 - At the end of the Abstract, it is necessary to address the importance and need for this study on a global scale.

2 - In the "Keywords" there are terms that are equal to the title, a more global idea of terms is suggested.

3 - The introduction is very well reasoned, but I suggest adding the importance of this study to the world at the end of the introduction. This would arouse the interest of readers more.

4 - The methodology is well founded. The authors really did a good job. Congratulations.

5 - The results are very well presented and discussed, the graphics are excellent and the conclusion is very well structured. Very good writing. This is remarkable.

6 - The Manuscript is well founded, together with the good quality of the English used in the text, which is clear and understandable. Congratulations.
